# The Biomodified Lignin Platform: A Review

**DOI:** 10.3390/polym15071694

**Published:** 2023-03-29

**Authors:** Filippo Fabbri, Sabrina Bischof, Sebastian Mayr, Sebastian Gritsch, Miguel Jimenez Bartolome, Nikolaus Schwaiger, Georg M. Guebitz, Renate Weiss

**Affiliations:** 1Department of Agrobiotechnology, Institute of Environmental Biotechnology, IFA-Tulln, University of Natural Resources and Life Sciences, Konrad Lorenz Strasse 20, 3430 Vienna, Austria; 2SAPPI Papier Holding GmbH, Brucker Strasse 21, 8101 Gratkorn, Austria; 3Austrian Centre for Industrial Biotechnology (ACIB), Konrad Lorenz Strasse 20, 3430 Vienna, Austria

**Keywords:** enzymes, technical lignins, circular economy, sustainable processes, green materials, enzymatic modification

## Abstract

A reliance on fossil fuel has led to the increased emission of greenhouse gases (GHGs). The excessive consumption of raw materials today makes the search for sustainable resources more pressing than ever. Technical lignins are mainly used in low-value applications such as heat and electricity generation. Green enzyme-based modifications of technical lignin have generated a number of functional lignin-based polymers, fillers, coatings, and many other applications and materials. These bio-modified technical lignins often display similar properties in terms of their durability and elasticity as fossil-based materials while also being biodegradable. Therefore, it is possible to replace a wide range of environmentally damaging materials with lignin-based ones. By researching publications from the last 20 years focusing on the latest findings utilizing databases, a comprehensive collection on this topic was crafted. This review summarizes the recent progress made in enzymatically modifying technical lignins utilizing laccases, peroxidases, and lipases. The underlying enzymatic reaction mechanisms and processes are being elucidated and the application possibilities discussed. In addition, the environmental assessment of novel technical lignin-based products as well as the developments, opportunities, and challenges are highlighted.

## 1. Introduction

The total European pulp production accounts for 4.4% of the sulphite pulp production, which results in 1.7 tons of available pulp. In general, most pulp production sites are in Sweden (31.2%) and Finland (30.2%). In total, 25.3% of the global pulp production is situated in Europe [1,2]. The worldwide production of technical lignin is approximately 100 million tonnes/year, valued at USD 732 million in 2015. It is expected to reach USD 913 million by 2025 with a compound annual growth rate (CAGR) of 2.2% [3]. Lignin represents the second most abundant source for sustainable aromatic polymers [4,5]. Although the structure of lignin is complex, it can be broken down into three repetitive structural motifs, the so-called monolignols. The complexity of lignin results from the many different linkages possible between the monolignol units as seen in Figure 1. These molecules are hydroxycinnamic alcohols containing a sidechain formed of three carbons (labelled α, β, and γ) attached to an aromatic ring system (labelled one to six), differing only in the number of methoxy groups attached on the aromatic ring. There is no methoxy group present in p-coumaryl alcohol, coniferyl alcohol has one at position C3, and sinapyl alcohol has two at positions C3 and C5.

While in nature, a cocktail of different enzymes (cellulases, cellobiohydrolases, peroxidases, and laccases) works synergistically on the degradation of lignocellulosic biomass, in the industry harsh conditions are required to separate biomass into its single compounds. In the paper industry, the pulping process generates purified cellulose and hemi-cellulose product streams. These are used for the generation of high-quality paper and fine chemicals. Meanwhile, lignin is regarded as a side product and is mainly burned for the re-generation of some of the energy needed during the pulping process. The rising awareness of the value of lignin, because of its structure, and in light of the circular bio economy concept value-added use of this resource is of more interest. Depending on the solvent used during pulping, different processes can be distinguished leading to different types of lignin with varying properties, the so-called technical lignins presented in Figure 2. The most common industrial processes are the sulfite, kraft, soda. and organosolv pulping [6,7,8].

In sulfite pulping, proper-sized wood chips (15–25 mm long) are cooked at high temperatures ranging from 140 to 170 °C at an acidic, neutral, or alkaline pH, depending on the sulfite salt added. The typically used counter ions are Ca^2+^, Na^+^, Mg^2+^, or NH^4+^. During cooking, the ether bonds within the lignin are hydrolyzed and subsequently sulfonated leading to the solubilization of lignin. The pulp not only contains lignin but also residual cellulose and hemicellulose, as well as some inorganic molecules. Thus, it is filtrated afterwards leading to accumulation of the lignin in the spent liquor. The modified lignin is then called lignosulfonate (LS), which is soluble in water (due to the sulfonation) of relatively high molecular weights (15,000–60,000 Da) when compared to other types of technical lignins and has a low concentration in phenolic groups [7,9].

In the kraft pulping process, wood chips are cooked for several hours at temperatures from 155 to 175 °C in an aqueous solution of NaOH and Na2S, the so-called white liquor. Under these extreme conditions the aromatic ether bonds crack, leading to the dissolution of the lignin and the precipitation of the cellulose and hemicellulose. This is also indicated by the color change of the liquor from white to dark brown or black. This black liquor has a high alkaline pH, typically around 13 to 14, rendering the lignin in its deprotonated form and thus soluble. In the black liquor, not only lignin is accumulated but also cellulose and hemicellulose residues, as well as inorganic compounds. Membrane filtration or acidification are methods applied to purify the kraft lignin (KL) further. In the industrially applied LignoForce process, membrane filtration is used. First, the black liquor is sparged with oxygen until the sulfite concentration is reduced to a specific level. Then, the solution is acidified by the addition of CO_2_ until a pH of 10 is reached. Finally, the lignin is separated by filtration and afterwards the filtration cake is washed with diluted sulfuric acid and dried, resulting in a high-quality technical lignin [10,11,12]. Another industrially applied purification method is based on acidification and is known as the LignoBoost process. Therein, the black liquor is first acidified with CO_2_ until a pH of around 10 is reached, resulting in precipitation of about 75% of lignin. The precipitated lignin is then re-suspended and further purified through the addition of H_2_SO_4_ until a pH of 3 is reached. To remove the remaining water-soluble parts of ash, washing with acidified water is necessary [10]. Upon the acidification of the black liquor, the deprotonated and thus soluble functional groups of lignin become protonated again (depending on their respective pKa values), leading to re-protonation, allowing for the formation of new linkages between the single molecules, leading to an increase in the molecular size, and, finally, to the precipitation [13]. This low-cost process results in kraft lignins of high purity and yield [14]. Kraft lignin generally is insoluble in water of low molecular weights (ranging from 200 to 20,000 Da) and has a relatively high concentration of phenolic groups [7,9].

Sulfite and kraft pulping are the industrially most common processes, with the latter being the dominant pulping process today. An example of a commercially available kraft lignin is Indulin AT. However, besides the structural changes of the native lignin molecule due to the harsh reaction conditions, sulfur groups are also incorporated, leading to severe alterations of the lignin molecule. In order to better understand the reaction mechanisms, it is essential to work with native lignin. In search for processes that allow for the isolation of more natural lignin, sulfur-free pulping processes were developed [15].

In the soda pulping process, mainly non-woody biomass is cooked at temperatures between 160 to 170 °C in the presence of NaOH and anthraquinone. The addition of anthraquinone is optional but leads to a higher lignin yield due to an increased ether bond cleavage. This process results in soda lignin, which is of a low molecular weight (800–3000 Da) and is sulfur-free and, thus, can be considered purer than other technical lignins, such as LS or KL [10,16]. The organosolv pulping process is another sulfur-free process. Various organic solvents, such as ethanol, methanol, acetone or mixtures of them with water, are used to dissolve lignin at high temperatures (100–250 °C) and pressures. The generated organosolv lignin a has high molecular weight (2000–9000 Da), is insoluble in water. and is closer to native lignin than other technical lignins (LS and KL). Organocell (using methanol as solvent) and Alcell (using ethanol as solvent) organosolv lignins are commercially available today [17]. Further, the hydrotropic delignification utilizing sodium benzoate is researched to obtain technical lignins with beneficial properties [18,19].

However, at the moment, there is no such thing as native lignin, as all technical treatments lead to at least slight modifications, such as condensations, of the lignin structure. Thus, new processes to isolate lignin are constantly developed, such as treatment with hot water, diluted acids, alkaline solutions, ionic liquid. or enzymatic hydrolyses. On a laboratory scale, milled wood lignin is thought to be the one that comes closest to native lignin. Thereby, ball-milled plant material is treated with a dioxane-water mixture as the solvent to extract the lignin from the plant cells [7].

From a bioeconomy perspective, lignin is currently mainly used to produce bioenergy (electricity and heat) but has recently received attention as a renewable raw material for the production of chemicals and materials to replace petrochemical resources and sometimes also provide technical improvements [20]. Other examples of interesting applications where lignin can be used to replace conventional materials are displacing urea-formaldehyde in adhesives [21], bitumen in asphalts [22], polyacrylonitrile in carbon fibers [23], and polyol in polyisocyanurate foams [20] and liquid fuels [24]. Yet, lignin is largely underexploited for these purposes, although many scientific studies are conducted to forward these fields [25]. Moreover, lignin can be used in other industrial applications that can benefit from the good surface activity of lignin [26], such as adsorbents for CO_2_ capture [27,28] and catalysts [2,29,30]. A number of chemical and physical processes for the improvement of the properties have been developed and reviewed in the past. This review, thus, mainly focus on the recent development of “green” enzyme-based upgrading of lignins.

When in nature the monolignols are incorporated into the growing polymer during enzyme-catalyzed lignification, the resulting phenylpropanoid units are called p-hydroxyphenyl (H), guaiacyl (G), and syringyl (S). The most common linkages formed between the monolignol units during radical cross-linking are aryl-ether (β-*O*-4), resinol (β-β), phenylcoumaran (β-5), biphenyl (5-5), and diaryl ether (4-*O*-5) linkages. Generally, ether linkages are more common than ester or carbon-carbon linkages. Amongst these linkages, the aryl-ether linkages (β-*O*-4) are the most abundant in native lignin, representing a relative labile linkage, easily cleaved during lignin pretreatment or depolymerization processes [4,6]. Aside from the interunit linkages present between the monolignols, their abundance and distribution also effect lignin reactivity. They vary vastly between different types of biomasses. Softwood lignin mainly contains G-units, in hardwoods both G- and S-units are present, and in grasses and herbaceous plants all three H-, G-, and S-units can be found. The structure of the monolignols also defines the possible structure of the formed polymers. G-units mainly build branched polymers due to the free position at C5, while S-units tend to form linear polymers. Hence, hardwood lignins are imagined as more linear polymers, while softwood lignins are expected to be rather branched [17].

In general, all technical lignins can be suitable for enzymatic modification or synthesis with the respective pretreatments. Kraft lignin, being readily available and of low cost, would be the best option from an economic and ecological point-of-view. Anyhow, the condensation of the C5 position in the aromatic ring, as well as the γ-elimination of the primary alcohol and the other conformational changes in enzymatic reactions are difficult to perform without extensive pretreatment. In addition, its low water solubility makes it necessary to engineer enzymes able to perform under extreme conditions such as high temperatures and at high pH values [31,32,33]. In contrast, lignosulfonates have been found to be more suitable for enzymatic modification. Their hydrophilicity due to the presence of anionic carboxylate groups, anionic sulfonate groups, and phenolic hydroxyl groups is beneficial. Furthermore, their polydisperse nature and wide range of molecular weight offer a multitude of reaction possibilities [34,35]. The lignins gained from soda pulping (also called organosolv lignins) are often described as the most native, and, therefore, often considered as beneficial for enzymatic modification. Studies show that enzymatic reactions are hindered by the low content of hydroxyl groups and the substitution of aromatic rings and steric barriers. Therefore, reactions like demethylations are employed [36,37].

With the latest geopolitical developments, the general availability and pricing of technical linins has vastly changed. Due to high energy prices, the burning of lignin for energy production stays in strong competition to their use as renewable raw material for the production of chemicals and materials to replace petrochemical resources. It is yet to be seen what consequence this has on further research and development in this area.

## 2. Methodology

The focus of this review is on enzyme-based strategies for the modification and valorization of technical lignins. Therefore, one of the first steps was to define the detailed content of the manuscript. Enzyme reaction mechanisms and technologies utilizing enzymes were structured as separate chapters. In light of the current sustainability discussion and environmental situation, we decided to introduce a chapter on the environmental assessment of existing processes and products. By utilizing databases such as PubMed (https://pubmed.ncbi.nlm.nih.gov/ (accessed on 8 February 2023)), the Web of Knowledge (https://www.webofscience.com/wos/woscc/basic-search, (accessed on 10 January 2023)), Scopus (https://www.scopus.com/search/form.uri?display=basic#basic (accessed on 15 January 2023)), and patent searches via the Austrian Patent Office (http://www.patentamt.at/ (accessed on 14 December 2022)), appropriate literature was analyzed in a comprehensive literature review. The responsibilities for the individual chapters were divided within the authors to optimize the research output. All authors reviewed all of the following chapters. The literature that was deemed too old or obsolete due to knowledge gain was excluded. The studies introduced were only eligible if they met the requirements of SCI journals and the guidelines of the University of Natural Resources and Life Sciences standards.

## 3. Enzymatic Modification of Technical Lignins

Various enzymes have been reported for their ability to modify technical lignin, namely laccases and peroxidases, both belonging to the oxidoreductases family (EC 1). Lately, another family of enzymes, namely hydrolases and, in particular, lipases, also have been investigated and found suitable for this modification, in particular for epoxidation (see Section 3.3). Many advances in this field are due to the changing legislation especially in the European Union involving the Green Deal. The introduction of environmentally friendly processes is a major part of this strategy [38]. In light of the circular economy, the use of waste and side stream materials is of great importance, with technical lignins showing great potential due to their beneficial properties described in the introduction. They are ideal candidates for enzymatic modification especially due to their structure and properties. With the need for a change in industrial processes and product lines, the use of enzymes is gaining traction, as most processes can be performed without organic solvents which often are toxic and harmful to the environment. In the following, the most promising enzymes are described [39].

### 3.1. Laccases

Laccases (EC 1.10.3.2) are multicopper-containing enzymes that catalyze the oxidation of various aromatic compounds, specifically phenols and anilines, while simultaneously reducing the molecular oxygen to water. In nature, this interesting group of enzymes is almost ubiquitous, since they can be found in higher plants, bacteria, fungi, and insects. In general, the natural role of laccases is related to their ability to polymerize or depolymerize a multitude of substrates: in higher plants, they are, together with peroxidases, involved in the lignification process and cell wall formation [40,41]; in fungi and bacteria they play a crucial role in lignin degradation and, in particular, act as virulence factors for many fungal diseases [42]; in insects, laccases have also been reported to be involved in the cuticle sclerotization process in the epidermis [43]. Moreover, their presence has been studied in soil and found to be responsible for the formation, among others, of humic substances [44,45]. Nevertheless, laccases have also been reported in many other natural processes such as pigment formation in fungal spores [46], iron metabolism in tulip trees [47], and the insect immune response [48], among others. The first laccase was discovered in 1883, when Yoshida demonstrated its presence in the Japanese lacquer tree *Rhus vernicifera* [49], and the first fungal laccase was found by Bertrand in a mushroom of the *Boletus* genus in 1896, which was responsible for the changing of the mushroom’s color when exposed to air. Since then, more than 60 laccase-producing fungi strains—mainly belonging to the class known as white rot fungi (i.e., basidiomycetes)—have been described, making fungal laccases the most representative group of the blue multicopper oxidase (MCO) family, concerning the number and extent of characterization [50,51]. On the other hand, bacterial laccases, which have not been extensively investigated yet, present peculiar properties compared to fungal laccases, such as a broad range of stability even at higher temperatures and pH values [52]. Furthermore, bacterial laccases are predominantly expressed intracellularly (in the form of inclusion bodies), while fungal laccases can be both intra- and extracellular. However, fungal laccases remain the first choice for many applications due to their high redox potential (E°) [53]. In fact, the redox potentials of laccases from common producing fungi are reported as 790 mV (*Trametes villosa*), 450 mV (*Myceliophthora thermophila*), 750 mV (*Pycnoporus cinnabarinus*), and 780 mV (*Botrytis cinerea*) [54], which are higher compared to the bacterial ones that are usually set at around 400 mV, such as *Streptomyces sviceus laccase* (wild type, 375 mV) or CotA laccase from *Bacillus subtilis* (455 mV) [55,56].

Laccase sequences can vary from 200 to 800 amino acids, leading to a molecular weight range of 50 to 140 kDa, including glycosylation which can contribute to up to 50% of the total weight [57]. Moreover, they have been reported to contain in some cases two or three cupredoxin-like domains, based on the organism they belong to [58]. Laccases exist in various forms, since they occur as monomeric or polymeric (homomeric, heteromeric, or multimeric) glycoproteins, with the majority of fungal laccases are identified as monomers, dimers, and tetramers. In all multicopper oxidases, the 3D-structure is well-conserved and consists commonly of beta sheets and turns; nevertheless, they can show peculiarities and differences regarding their catalytic properties, sequences, and post-translational modifications [58]. The active site of the laccases shows four copper atoms, which mediate the redox process. These redox sites are typically classified depending on their role in the catalytic cycle: Type1 (T1) or “blue-copper center”, Type2 (T2) or “normal site”, and Type3 (T3) or the “coupled binuclear copper center”. Specifically, Type1 copper is coordinated with one methionine, one cysteine, and two histidine residues, giving the classic blue intense color of laccases and showing a strong electronic absorbance at 610 nm [59]. Type2 copper is colorless, not detectable by adsorption, and coordinated with two histidines and a water molecule. Both Type3 copper atoms are coordinated with three histidine residues (for a total of six), presenting an anti-ferromagnetic coupling and a hydroxyl bridge between the copper pair (Figure 3). This copper pair shows a weak absorbance at around 330nm [60]. In each reaction cycle, one electron is transferred from the substrate to Type1 copper which is thus responsible for its oxidation. Because of this, the laccase redox potential is determined mainly by the T1 copper center. On the other hand, T2 and T3 coppers form a trinuclear cluster (TNC) that catalyzes the reduction of oxygen to water requiring four electrons. Thus, electrons resulting from fours cycle of substrate oxidation by the T1 blue-copper center are transferred to the trinuclear cluster where O_2_ is reduced, yielding two molecules of water. The different hydrogen bonds and salt bridges that are present between the copper atoms provide enzyme stability. Since all four Cu atoms are fully oxidized (Cu^2+^) in the native form of the enzyme, laccases are able to catalyze the decarboxylation, demethoxylation, and demethylation of phenolic compounds, which are fundamental steps in lignin depolymerization [61]. Furthermore, the radical species produced through the catalytic mechanism enable laccases to offer both synthetic and degradative processes. In fact, as classified by Pezzella et al., those species can lead to different pathways depending on the substrates and reaction conditions: (a) oxidative coupling, (b) dead-end products, or (c) bond cleavage [62]. Oxidative coupling involves homo- or cross-coupling that leads to the formation of dimers or polymer species. Lignin polymerization and grafting for surface modification are the most important examples. On the other side, dead-end products occur depending on the stability and redox reversibility of the generated radical species, which can undergo a rearrangement per se or oxidize another compound that acts as a mediator to further oxidize another non-phenolic compound causing bond cleavage [63]. The biotechnological applications of these two pathways involve bioremediation and bleaching technologies, for instance [62].

Generally, laccases can oxidize an exceptional range of natural substrates (i.e., polyphenols, benzenethiols, aromatic or aliphatic amines, and many others). Even molecules showing a high redox potential (E° > 1100 mV) or steric hindrance such as non-phenolic compounds, which normally cannot be oxidized directly by the laccases, are able to undergo the oxidation reaction thanks to the laccase mediator system (LMS), formed by small molecules called “chemical mediators” that can interact with the enzyme [64,65]. In this way, the mediators act as an electron carrier between the laccase and the substrate to be oxidized. Therefore, as already mentioned above, LMSs play a key role in the depolymerization of lignin. More than 100 compounds have been reported as laccase mediators and are usually divided into natural and synthetic mediators [57,66]. Furthermore, 2,2′-azinobis(3-ethylbenzthiazoline-6-sulfonate) (ABTS), 2,2,6,6-tetramethyl-1-piperidinyloxyl (TEMPO), N-heterocycles bearing N–OH groups (and in particular 1-hydroxybenzotriazole, HBT), violuric acid (VLA), N-hydroxyphthalimide (HPI), and N-hydroxyacetanilide (NHA) are all examples of artificial mediators. On the other hand, syringaldehyde, acetosyringone, vanillin, acetovanillone, methyl vanillate, ferulic acid, sinapic acid, *p*-coumaric acid, and others are all lignin-derived mediators which have been reported and investigated as natural mediators to replace the synthetic ones [67]. An ideal mediator should be non-toxic, economic, and efficient, with stable oxidized and reduced forms that do not inhibit the enzymatic reaction at hand [68]. Further information regarding the LMS and the latest results obtained with technical lignin are discussed in Section 3 (Enzymatic depolymerization and delignification). Further, the immobilization of laccases on nanoparticles can improve its properties. In a story by Chen et al., immobilized laccase showed improved thermal stability and pH tolerance compared with free laccase, and more than 80% of its initial activity remained after 20 days of storage at 4 °C. In addition, about 40% residual activity of the laccase remained after 8 cycles [69]. Due to the extensive biotechnological studies and their already wide applications, for instance, in the pulp and paper industry [70,71], in the textile industry [62,72], or in the bioremediation and detoxification of many industrial processes [73,74], laccases cover a central role in the modification of technical lignin [75].

### 3.2. Peroxidases

Peroxidases (EC 1.11.1.7) are enzymes that are able to interact with various peroxide species (ROOH, such as hydrogen peroxide) as electron acceptors to catalyze a wide number of oxidative reactions, reacting with organic and inorganic substrates. Similar to laccases, peroxidases are ubiquitous and abundant in nature in all forms of life, since they can be found in plants [76], bacteria, fungi, and even humans (e.g., glutathione peroxidase) [77]. Peroxidases can be broadly categorized as heme or non-heme peroxidases, depending on the presence of the prosthetic group. Generally, based on their origin, heme-peroxidases can be divided into two different superfamilies, which are the superfamily of Plant, Fungal, and Bacterial peroxidases and the superfamily of Mammalian and other Animal peroxidases [78]. Examples of the first superfamily are cytochrome C peroxidase (CCP, prokaryotic), lignin peroxidase (LiP, basidiomycete), and horseradish peroxidase (HRP, plant), while lactoperoxidase (LPO) and myeloperoxidase (MPO) are part of the second superfamily [78]. In addition, four small families belong to this classification since they are able to reduce peroxides using a heme group: (1) Catalases, (2) Di-heme cytochrome C peroxidases, (3) Dyp-type peroxidases, and (4) heme halo-peroxidases. On the other hand, non-heme peroxidases, which interact with peroxides species without the presence of the heme group, are independent from an evolutionary point-of-view and are divided into five families, based on the molecule they use in their active site. The largest family is represented by thiol peroxidases, which are divided into glutathione peroxidases and peroxiredoxines, while the other families are alkyl-hydro peroxidases, non-heme halo peroxidases, manganese catalases, and NADH-dependent peroxidases [79]. In this paragraph, we focus on heme peroxidases, accounting for the majority of all of the peroxidases (>80% based on the PeroxiBase database for gene coding [76,80]) and are the most promising for industrial applications and technical lignin modifications. Peroxidases range from 251 to 726 amino acids, providing a molecular weight that varies from 35 to 150 kDa [81,82]. The superfamily of Plant, Fungal, and Bacterial peroxidases is formed of monomeric proteins, while the one of Mammalian and other animal peroxidases shows alpha-helical monomeric and dimeric glycosylated enzymes. Their structure is conserved and usually consists of 10-11 alpha helices, which are linked by loops and turns, while the presence of β-structures is generally a minor component or even absent [83]. The catalytic center of peroxidases consists of iron protoporphyrin IX, the prosthetic heme group showing Fe (III) in the resting state. Moreover, an important role is played by histidine (His) as the proximal iron ligand and a water molecule in the distal side of the pocket that is not coordinated by the iron center. In particular, these structural features are considered to be responsible for the specific biological function of the enzyme, the reduction potential of the iron (together with the protein matrix around the prosthetic group), and the nature of the substrates which can be oxidized [84]. Furthermore, a distal histidine not connected to the iron atom was reported to be closely involved in the catalysis [85]. The first step of the catalytic cycle consists of the binding of the ferric atom in its resting state (III) and the peroxide, usually hydrogen peroxide, to form the so-called “Compound 0” (ferric hydroperoxide intermediate). The next step involves the transformation of “Compound 0” to “Compound 1”, which is possible due to the protonation of the distal oxygen of the complex. In this way, the molecule of water is eliminated and “Compound 1” is formed, characterized by a ferryl species [Fe (IV)=O)], coupled with a porphyrin radical cation. Once the radical cation gets extinguished by one-electron reduction (donated by a second molecule of oxygen peroxide or by the substrate), “Compound 1” forms “Compound 2”, in which the iron atom is still present in the [Fe (IV)] state. Finally, “Compound 3” is established and consists of a complex with the iron atom in the ferrous state (II) bound to the superoxide ion (O^2•–^). “Compound 3” is most likely formed through a large excess of hydrogen peroxide (H_2_O_2_). After catalyzing the one-electron oxidation of a substrate molecule, “Compound 3” returns to the resting state of the enzyme and one catalytic cycle is considered finished [78,86,87]. Figure 4 visualizes the general catalytic cycle of heme peroxidases. All of the reactions catalyzed by peroxidases can be divided into four different types, following the classification by Vamos-Vigyazo and Haard [87]: peroxidative, oxidative, hydroxylation, and catalytic. In particular, as mentioned before, the structural features of the protein and the structural arrangements determine the specificity towards different substrates. Regarding the different types of reactions, this is determined mainly by the nature of the substrate and, ultimately, by the enzyme properties.

The wide range of substrates that can be accommodated by peroxidases makes these enzymes suitable and useful for a large number of biotechnological applications. Lignin peroxidases (LiPs) and manganese peroxidases (MnPs) have been extensively studied for their lignin degradation activity and they may be successfully utilized in the paper industry, especially for bioleaching and bio-pulp purposes, in which they could offer an effective alternative to chemical and mechanical pulping [88,89]. Furthermore, the same two enzymes are usually part of the “lignocellulosic cocktails” used for the enzymatic treatment of lignocellulosic fibers for the production of second-generation biofuels [90]. Moreover, potential applications for dye decolorization and the treatment of textile effluents have been reported using peroxidases [91,92]. On the other hand, regarding environmental efforts, peroxidases and, in particular, horseradish peroxidase (HRP) have been studied in soil detoxification [93,94,95] and the bioremediation of wastewaters contaminated with phenols and derivates, among others [94,95]. Biosensors utilizing peroxidases have found application in the quantification of hydrogen peroxide, among others, while diagnostic kits for analytical applications are already in use for the determination of glucose, lactose, uric acid, and others [93,96,97]. Peroxidases are also used to detect the proper execution of heat treatment; for instance, in dairy products and UHT milk (ultra-high temperature), to verify that the sterilization has been carried out properly, peroxidase tests are performed and enzyme activity must be absent [98]. Lignin nanoparticles are used as a renewable and efficient platform for the immobilization of horseradish peroxidase and glucose oxidase. Cationic lignin performs best as a polyelectrolyte in the retention of the optimal Con A aggregation state. Electrochemical properties, temperature and pH stability, and reusability were found to be better than, or comparable to, that previously reported for other HRP–GOX immobilized systems [99].

### 3.3. Lipases

Lipases (triacylglycerol acyl hydrolases, EC 3.1.1.3) are enzymes of the “α/β-hydrolase fold” family, having a structure composed of parallel β strands that are surrounded by α helices [100]. They do not require cofactors and belong to the class of serine hydrolases [101,102]. These enzymes catalyze the hydrolysis of long-chain triglycerides, releasing free fatty acids, mono- and diacylglycerols, and glycerol. In nature, these enzymes are ubiquitously distributed among a wide variety of organisms (higher animals, microorganisms, and plants), which use them for metabolizing different lipids. Lipases have a conserved pentapeptide sequence (Gly-X-Ser-X-Gly) called the “nucleophilic elbow”, since said sequence forms a β-turn-α motif [103]. They have an active site composed of a catalytic triad of Ser-Asp/Glu-His residues. This active site is also covered by an α helix peptide loop, acting as a lid, that can be reoriented at the interfaces of biphasic systems (i.e., interfacial activation) [104], although more recent studies suggest a more complex behavior [105]. This lid closes when the enzyme is in an aqueous media, blocking the hydrophobic catalytic center [106]. The lipases catalyze esterification, interesterification, and transesterification reactions when they are solubilized in an organic medium [107]. When they are solubilized in a mixture of water and organic solvent, lipases catalyze the hydrolysis of carboxylate ester bonds into free fatty acids as well as organic alcohols, at the organic-aqueous interface [108]. For their catalytic mechanism, a nucleophilic attack of the carbonyl group of the substrate is conducted by the serine’s hydroxyl residue. This results in the formation of an acyl-intermediate that is deacetylated by a different nucleophile, resulting in the release of the product and in the regeneration of the catalytic site [109].

Lipase-catalyzed reactions with lignin have been scarcely studied, although the epoxidation of alkenes using lipases in the presence of free fatty acids has been known for a long time (Figure 5) [110]. Recently, this reaction has been studied on lignin model compounds (i.e., vanillic and gallic acid) [111], as well as in organosolv lignin [112], in the presence of caprylic acid. Here, epoxides were formed from lignin and its model compounds after they were allylated.

Lignin model compounds have also been successfully esterified using lipase B form *Candida antartica* (CALB), which could potentially lead to an increase in the different properties of lignin, such as hydrophobicity or antioxidant activities [113]. CALB has been used for the delignification of lignocellulosic biomass dearomatizing lignin oil. This can be performed thanks to the production of peracids formed in situ by the lipase [114].

Interestingly, lignin can also be used to immobilize lipases. Lipases have low stabilities and are very sensitive to environmental conditions, such as suboptimal pH values. This makes the immobilization of the enzyme extremely important, especially since a large number of industrial applications have harsh reaction conditions that lipases would not withstand by themselves [115]. Lignin has been used as a way to enhance cellulose microspheres [116,117] and to produce magnetic lignin composites with iron oxide nanoparticles [118], using a chitosan layer covering the lignin nanoparticle [119] or mixed with choline chloride [120].

## 4. Techniques/Processes for the Valorization of Technical Lignins

Since the awareness of the value that lies within technical lignin has grown, a lot of research on lignin itself and possible processes to obtain valuable lignin products has been conducted. Many lignin modification processes have been developed since then, following different strategies (Figure 6). These strategies can be classified into three main approaches (1) the depolymerization of lignin followed by the synthesis of value-added chemicals; (2) the modification of lignin to obtain a higher reactivity or different properties; and (3) further polymerization of lignin. Generally, modification of lignin can be performed chemically, thermally, or enzymatically [7,121,122,123].

The most common thermal and chemical processes to obtain valuable lignin products are pyrolysis (thermolysis), gasification, hydrolysis (under supercritical conditions), reduction (hydrogenolysis), or oxidation. All of these processes are aimed at the depolymerization of lignin to obtain low molecular weight phenolic compounds, carboxylic acids, or other aromatic compounds. These compounds can either be the desired products themselves (vanillin) or they are used as starting material for the synthesis of further fine or bulk chemicals (syringaldehyde) [124,125]. In pyrolysis, lignocellulosic biomass is degraded at high temperatures (up to 600 °C) and under a limited amount of oxygen, resulting in solid (tar, char, and coke), liquid (phenols, ketones, aldehydes, alcohols, carboxylic acids, esters, ethers, furans, sugars, alkenes, nitrogen, and oxygen compounds), and gaseous (carbon monoxide, carbon dioxide, hydrogen, ethane, methane, propane, sulphur oxides, nitrogen oxides ammonia, and ethylene) products. The temperature and residence time chosen for the process influences the yield and properties of the resulting product. A low temperature and residence time mainly produced charcoal, while at moderate temperatures and low residence times the formation of liquid products is favored. The selectivity of this process is rather low and thus cost-intensive separation processes have to be employed afterwards [126,127,128]. The acid hydrolysis of lignocellulosic biomass can be used to generate second-generation biofuels, cellulose nanocrystals, lignin hydrogels, and adhesives. The strength of the acid used can vary from mild to strong acidity and influences the quantity of the products, while the reaction time did not seem to be crucial in that regard. C5 sugars (hemicellulose) are more effected by severe acid hydrolysis than C6 sugars (glucose) [129]. Although O_2_ is widely used as an oxidant for the processing of technical lignins to phenolic compounds, other catalysts show higher oxidative power. The chemical catalysts used for the oxidative depolymerization of lignin can be metal salts, organometal compounds, metal-free organo-catalysts or acids, and bases. The type of catalyst and the reaction conditions such as pH, temperature, or the surrounding atmosphere determine which part of lignin is oxidized and thus influence which product accumulates in the end. When the process is run with either nitrobenzene or alkaline oxidation at 120 °C, an oxygen partial pressure around 3 bar and a 60 g/L lignin solution prepared in a 2 N NaOH solution, a good balance of enhanced conversion of phenolics and minimizing oxidation was achieved. However, depending on the source of the biomass and the delignification process used, the outcome of the reaction and the residual products formed may vary [125,130]. However, in general, chemical processes use harsh reaction conditions (pressure, temperature and pH) and harmful chemicals, such as organic solvents, strong alkaline, or acidic milieus, and lead to the accumulation of environmentally harmful waste.

Enzymatic processes, on the contrary, can be run at lower temperatures, leading to a lower energy demand, the need for less harmful chemicals, and the accumulation of less waste. For these reasons, the use of enzymes in industrial processes is regarded as beneficial. In the case of lignin modification, the most-used enzymes are peroxidases and laccases. Laccases are preferred to peroxidases, since they require only molecular oxygen (O_2_) as a co-substrate, while peroxidases need hydrogen peroxide (H_2_O_2_) as a co-substrate, which is known to be destabilizing to enzymes and further leads to higher costs of the processes [51,131,132]. At the moment, a lot of research is going on to find possible applications of the laccase catalyzed oxidation of technical lignins in the industry. Since laccases either can act in the synthesis or in the degradation of lignin and various technical lignins differing in their properties are available, those processes need to be adapted accordingly to obtain the desired products. The factors influencing the reaction are the source of the biomass (hardwood, softwood, grasses), the delignification process (lignosulfonate, kraft lignin, organosolv lignin, soda lignin, milled wood lignin, etc.), the choice of laccase (high or low redox potential, bacterial or fungal origin), and the use of mediators and the reaction conditions, such as pH, temperature, or type of solvent. The reaction temperature affects the laccase-lignin reaction as it influences both the catalytic activity of the laccase as well as the solubility of lignin [133,134]. In terms of pH, the incubation of lignosulfonate with ThL (a high redox potential laccase from the fungus *Trametes hirsuta*) was found to deliver better results at pH 6 to 7 than at pH 3 to 5, where fungal laccases normally have their optimum activity [135]. In another study, it was found that the dosage of enzyme used for the reaction also had a strong influence on the outcome of the polymerization; a higher dosage of MtL (a low-redox potential laccase isolated from the fungus *Myceliophthora thermophila*) was found to increase the degree of polymerization [7]). Later on, it was found that this effect depended on the specific enzyme, since this effect was not found when a high-redox potential laccase from the fungus *Trametes villosa* (TviL) was used. In the same study, it was shown that higher concentrations of lignosulfonate lead to higher molecular weights, independent of the laccase used [7,136]. On the other hand, this effect was not observed for other technical lignins, such as kraft lignin, upon incubation with an immobilized MtL laccase [137]. Several studies aimed at the enzymatic modification of various kraft lignins (softwood, hardwood, grasses) with laccases of different origins (fungal or bacterial) with and without the use of mediators. It turned out that diverse laccases are able to polymerize kraft lignin both under slightly acidic or alkaline conditions, depending on the laccase used, and that the use of mediators is not necessary to obtain better results [138,139,140,141]. A recent study also found that the method of drying of the purified lignin influences the reactivity. Therein, oven-dried lignin was compared to spray--dried lignin; spray-dried lignin showed improved solubility and better polymerization abilities than the oven-dried one. It was assumed that the smaller, more regularly distributed and the more uniform surface topography of the spray-dried lignin particles were the reasons for this observation [142]. Another possibility to enhance lignin polymerization by laccase is the addition of mediators. Extensive polymerization of lignosulfonate was achieved by the use of ThL or TviL laccase in combination with 1-hydroxybenzotriazole (HBT), caffeic acid, or vanillyl alcohol as mediators [143,144]. The polymerization of an organosolv lignin at pH 10 was sown by the combination of a bacterial laccase (SiL) and acetosyringone as the mediator [123,138]. Many other laccase mediator systems are known, but most of them are aimed at the depolymerization of lignin (see Section 3). Aside from the polymerization and depolymerization of lignin, the radical reaction mechanism of laccase with lignin allows for the grafting of foreign molecules onto lignocellulosic biomass or technical lignins. With this approach, the properties of lignin can be tailored into a wanted direction by choosing the respective type of technical lignin, laccase, solvent, and functional molecule. It was shown that acrylic compounds, water-soluble and plant-based phenols, nitrogen-containing compounds, inorganic silanes, cellulose, and thiols could be grafted to technical lignins. This allows for potential applications in lignin-based polymers (plastics), thermoplastics, adhesives, or new materials. With this approach, a new type of technical lignin, namely Ecohelix, was discovered. It was generated by coupling hemicellulose with aromatic moieties (such as lignin residues) by laccase. This new technical lignin has high sulfonate and aliphatic OH-groups and a suitable molecular weight and thermal stability, which can probably be used for the generation of new materials, such as films [145]. Further, with this process, water-insoluble lignins were turned into water-soluble lignins by coupling lignin to low-molar-mass organic compounds. Furthermore, lignins with amphiphilic properties were generated by laccase-catalyzed cross-linking of alkali lignin in the presence of epichlorohydrin, a sulfonating agent, and an aldehyde compound [7]. A laccase-catalyzed process for the extensive polymerization of lignosulfonates was implemented (Figure 7). This process only needs lignosulfonate dissolved in water, set to a pH of seven with NaOH, a steady external oxygen supply and a laccase. For lignosulfonate, the laccase of the thermophilic fungus *Myceliophthora thermophila* (MtL) was used, resulting in extensively polymerized lignosulfonate, which showed insolubility in water and better dispersing abilities [143,146]. These newly generated properties allow for possible applications of this process for the production of hydrogels, coatings, adhesives, and biopolymers for use in agriculture, paper, board, wood, and textile gluing [147,148,149].

### 4.1. Plasticizers

Enzymatically polymerized lignosulfonates produce water-insoluble materials with the potential of being cast into any shape [146]. These polymers are highly brittle once they dry, which prevents them for being further processed into materials. These insoluble lignin materials could have their mechanical properties improved using plasticizers, a possibility known since 1975 [151]. These compounds are low in molecular weight, non-volatile molecules widely used as additives for enhancing the flexibility and processability of different materials. The plasticizers occupy the intermolecular spaces between polymer chains, which reduces the secondary forces among them. This changes the three-dimensional molecular organization of polymers, reducing the energy that is required for molecular motion and promotes the formation of hydrogen bonding between the different polymeric chains [152]. These compounds can lower the second order transition temperature, the glass transition temperature (Tg), density, hardness, electrostatic charge, and viscosity of the polymers, increasing the polymer’s resistance to fracture by avoiding the formation of cracks in their matrix [153]. The effects of plasticizers depend on the chemical structure, molecular weight, and functional groups they have, giving a variety of behaviors to the lignin depending on the additives used [153]. Recently, it was found that the addition of glycerol to laccase-polymerized lignosulfonate increased its elongation at break 111%, while the addition of Poly(ethylene glycol) (PEG) 600 increased the tensile strength to 74 MPa [154]. Lignosulfonates were blended with polyvinylpyrrolidone (PVP) and zeolite, improving the characteristics of the material [155]. Figure 8 shows the effect of various plasticizers on the tensile strength and elongation break of enzymatically polymerized lignosulfonates [154].

The addition of acetyl tributyl citrate (ATBC) to lignin-based polyurethane films in another study increased their tensile strength to 16.68 MPa and the elongation at break to 246.36% [156]. For kraft lignin, PEG was also used as a plasticizer, determining that the plasticization efficiency increases with the number of oxygen atoms of their chains, as well as the optimum kind of plasticizers used depending on if the lignin is dry (using hydrogen bond forming molecules) or solubilized (using aromatic molecules such as vanillin) [157]. Kraft lignin, as well as organosolv lignin, has also been plasticized using different poly(vinyl chlorides) (PVC) such as diethylene glycol dibenzoate (Benzoflex 2-45), tricresyl phosphate (Lindol), or alkyl sulfonic phenyl ester (Mesamoll) [158].

### 4.2. Hydrophobicity

The increase in the hydrophobicity of technical lignins could be utilized to improve the performance of lignin-based materials by expanding their range of applications. Enzymes have been assessed to covalently incorporate hydrophobic molecules into lignins while fluorophenol molecules (FPs) were used as models, especially 4-[4-(trifluoromethyl)phenoxy]phenol (4,4-F3MPP). FPs have been successfully grafted on lignin model substrates [159] and lignocellulose fibers [160,161] and onto isolated lignosulfonates [150]. In the latest experiment, the water contact angle of the polymerized lignosulfonate with FP as an additive increased by 59.2% and the water swelling decreased by 216.8%, showing the ideality of the use of FPs as additives for improving the wet resistance. On the other hand, contrary to the lignosulfonates previously mentioned, kraft lignins already show a high hydrophobicity without needing to use additives [162]. This leads to the kraft lignins being used as additives themselves to improve the hydrophobicity of other materials. For example, kraft lignin has been used for improving the hydrophobicity of thermoplastic starch (TPS), increasing their water contact angles from 65.52° with pure TPS to up to 90.25° when 8% of kraft lignin was present [163] and decreasing their water absorption from 23.81% to 18.60% when kraft lignin was present [164].

### 4.3. Epoxidation

The addition of epoxide groups to the lignin, using compounds such as epichlorohydrin, is a well-known strategy to improve the properties of lignin-based materials. Complementing what has been said in the previous section, the increase in both the water and organic solvent solubility of the lignin can also be interesting for developing industrially valuable materials such as surfactants. The addition of PEG (a plasticizer already mentioned in Section 4.1) to the alkali lignin previously epoxidized with epichlorohydrin resulted in a non-toxic water-soluble material with the potential for being used as detergents or emulsifiers [165]. The epoxidized lignin also obtained a lipophilic capacity, making it soluble in a variety of common organic solvents, such as methylene chloride, acetone, chloroform [166] or tetrahydrofuran [167]. The addition of epichlorohydrin to the lignin not only improves the solubility, but, recently, this additive has been used for improving the mechanical properties of epoxy adhesives, increasing the shear and tensile strength, adhesion strength, or improving thermal stability [168,169].

### 4.4. Enzymatic Depolymerization and Delignification

As previously mentioned, lignin is a fundamental part of the plant biomass, specifically in the lignocellulosic biomass. This is a green and renewable source for carbon, opposed to the petroleum-based sources. One of the main issues with lignocellulose is its recalcitrance, making it necessary to separate its different constituting elements for obtaining value-added products [170]. The delignification is a necessary technique for different industries such as the production of bioethanol or in the pulp and paper industry [171,172]. Lignin is closely associated with the cellulose and hemicellulose, a source of polysaccharides, preventing the access of hydrolytic agents that could convert them into easier-to-use monosaccharides, unless it is removed by chemical or biological methods [173]. In this review, we focus on the second kind, more specifically in the use of enzymes. In nature, the lignin removal is mainly conducted by the so-called “white-rot fungi”, which are part of the basidiomycetes division, although some bacteria can also be involved in the process [174,175]. They use a wide array of oxidative enzymes, mainly ligninolytic peroxidases from the class II of the peroxidase-catalase superfamily, for the extracellular degradation of the lignin [176,177]. These enzymes, which have been previously described in Section 3.2 [78], are classified into three families according to their oxidation site’s molecular architecture: MnP, whose Mn-binding site is used for oxidizing Mn^2+^ to Mn^3+^, oxidizing the phenolic moieties of lignin [89,178]; LiPs, with a catalytic tryptophan that can directly oxidize high-redox potential aromatic substrates that are non-phenolic [88,179]; and versatile peroxidases (VPs), that have the catalytic sites of both LiPs and MnPs combined into one enzyme [180]. From all of these, the LiPs are the most efficient in terms of oxidizing the lignin [176,181]

## 5. Applications

### 5.1. Coatings

Coatings are covering layers applied to the surface of a substrate which can serve decorative, functional, or both purposes. They are often applied to surfaces to increase the mechanical resistance as well as protect against environmental influences and corrosion (Figure 9) [182]. The inherent structure of lignin yields some very favorable attributes such as UV absorbance, antioxidant, and antibacterial properties which make lignin an interesting material for coatings [183,184,185]. Kaur et al. have reported good UV absorbance and antimicrobial properties against Gram negative bacteria of glass slides and cotton fabrics coated with lignin-based coatings, all while maintaining clear and transparent properties. These effects were further increased when the coating was doped with lignin-titanium-dioxide nanoparticles [186]. Aqueous solutions of lignosulfonate, alkali lignin, and enzymatic hydrolysis lignin have been used as flame-retardant coatings for polyurethane (PU) foam, as reported by Zhang et al. [187]. While uncoated PU foam quickly burned after an initial flame, leaving little residues, lignosulfonate- and alkali-lignin formed a protective char barrier, preventing the fire from spreading and ultimately quenching the flame. Zhang et al. reported higher protective effects with increasing lignin concentrations. Only enzymatically hydrolyzed lignin showed poor flame-retardancy as the residues of cellulose and hemicellulose seemed to counteract the protective effects. Conclusively, an increase in the limiting oxygen index (LOI: the minimum oxygen concentration that supports the combustion of a material) from 17% for uncoated PU to 28% for coated PU was recorded. While unmodified lignin was used in these examples, many times lignin is modified to enhance its properties or introduce new functionalities. A hydrophobic thermosetting polyester of succinated kraft lignin coating showed improved scratch resistance and hardness compared to unmodified lignin and proofed excellent adhesion to wood and glass, as well as comparable results to fossil-based polyester coatings on aluminum [188]. Haro et al. reported improved hydrophobic characteristics and corrosion resistance comparable to conventional passivated aluminum when spin-coating the metal with cross-linked silanized kraft lignin [189]. Similarly, Cao et al. produced a lignin-based polyurethane coating polymerizing lignin-polyols with hexamethylene diisocyanate. These coatings were applied to carbon steel to improve the corrosion resistance and displayed high mechanical strength and thermal stability [190].

Esterification is also a popular modification. Through the grafting of fatty acids to lignin hydroxyl groups, the hydrophobicity can be drastically increased. Hult et al. applied a coating based on kraft lignin esterified with tall oil fatty acids to paper boards using a meter bar coater which significantly reduced the water vapor and oxygen transmission rates without influencing the tensile strength of the paper itself [191]. In a similar study, Hua et al. spray- and spin-coated a dispersion of kraft lignin esterified with oleic acid onto glass, wood, and pulp sheets, increasing the hydrophobicity of all materials as illustrated by a rise in the water contact angles [192]. Most of these lignins were modified chemically. However, as mentioned in Section 3.3, the enzymatic esterification of lignins has already been demonstrated, pointing towards a more environmentally friendly future for the preparation of such coatings [113].

In agriculture, lignin coatings have been reported for a long time. In 1996, Garcia et al. first examined the use of lignin as a coating for urea pellets [193]. Since then, lignin-based slow-release coatings for fertilizers have been widely discussed [194,195,196,197,198]. Most of these coatings exploit the natural hydrophobicity of kraft lignin to reduce the solubility of the coated fertilizer, thereby providing a gradual, long-lasting fertilizer supply for the agricultural crops. A different approach was chosen by Weiß et al., who produced a coating based on enzymatically polymerized lignosulfonate, serving as carrier for plant growth promoting microorganisms and significantly reduced substrate abrasion [199].

Lastly, spin-coating is also used to produce lignin model surfaces which are necessary for adsorption and surface interaction studies and are, therefore, an important tool for future research developments [200,201]. Thus, so far untested molecules, such as enzymes, could be immobilized on lignin surfaces, opening up possibilities for new functional lignin coatings.

### 5.2. Adhesives

Lignin is already known as nature’s ‘glue’ in wood and other higher plants [202], and is attracting the interest of both pulp producers and material scientists since lignin features special properties including aromaticity, antioxidant, and antimicrobial activity [203]. Lignin has two different hydroxyl groups: aliphatic and phenolic, the latter being among the reactive functional groups that can be altered [204]. One of the reactions that lignin can undergo is phenolation, condensing phenol with aromatic rings and, consequently, incorporating more phenolic hydroxyl groups. This increases the potential of lignin as a phenol binder, being a valid alternative for conventional binders such as phenol-formaldehyde (Figure 10) [205].

Modified lignin in use for adhesives, e.g., using paper boards in glue-promoting interlayers, covering concrete floors with linoleum bonding [206] have been filed in respective patents since the 1950s. Concerning special applications, the blending of lignosulfonates with different additives, i.e., Cr(VI) salts and gypsum for bonding of floor coverings [207] or dextrins for papers glues, were investigated. More recently, glues based on mixtures of polysaccharides (potato starch) or monosaccharides (sorbitol) combined with magnesium lignosulfonate have been proposed [208]. Lignin also can be used in combination with, or even completely replacing, traditional phenol-formaldehyde adhesives, which reduces the use of toxic compounds while, at the same time, maintaining the properties of the adhesives [209,210].

Lignin has not only been used as a means to improve the properties of other materials as a binder, but also as the main component of adhesives. Technical lignins can be employed both on their own or in combination with other polymers. As recent examples of adhesives using solely lignin, in this case oxidized-demethylated lignin with an addition of 20% NaIO_4_ was used as a bio-based wood adhesive, obtaining an acceptable shear strength even in wet conditions [211]. The use of colloidal lignin particles [212] as a bio-based adhesive was recently investigated, obtaining a pure lignin adhesive with eight times higher shear strength than the tensile strength [213].

On the other hand, as examples of adhesives using lignin in combination with other polymers, chitosan was combined with ammonium lignosulfonate, interacting between them by hydrogen bonding, to produce a wood adhesive [214]. Tannins have also been used in combination with lignin for developing adhesives for fiberboard manufacturing, obtaining materials that satisfied the European standards [215].

The previous adhesives examples use kraft lignin or lignosulfonates that are unpolymerized. Enzymatically polymerized lignosulfonates have also been used for developing adhesives. One of recent discoveries is the technique to convert lignosulfonate enzymatically into wood adhesives to fully bio-based composite materials [149]. Here, the results showed that laccase-polymerized lignosulfonate-based wood adhesives had comparable characteristics as the commercially available glues in the sector of D3 class white glues (PVAc-based). No significant difference was monitored regarding the tensile shear strength between these two types of adhesives.

Although, as previously mentioned, lignin has extensively been used as a wood adhesive, there are recent examples of it being used in adhesives for different adherends. For example, the addition of enzymatically polymerized lignosulfonates to starch-based paper adhesives has proven to be a successful strategy for improving their properties, mainly their wet resistance, increasing it from 150 min when only a starch adhesive was used to 1200 min when a polymerized lignin was added [216]. Recently, there has been increased interest in using lignin-based binders for lithium-ion batteries. Lignin has been used as a binder, polymer electrolyte, and an electrode material for organic composite electrodes/hybrid lignin-polymer combination in different battery systems depending on the principal charge of quinone and hydroquinone. Lithium-ion batteries are a relatively novel technology that are used in high-end electronics. They differ from more classical batteries in the fact that for lithium batteries the electrolyte is not consumed. Here, the same ion participates at both electrodes, with the possibility of being reversibly inserted (and extracted) from the electrode material [217,218]. Lignin has been successfully utilized as a greener alternative for the conventional polymer binders [219,220].

### 5.3. Fillers

The low value and abundance of the technical lignins make it perfect for using it as a filler for polymers [221]; this could be useful for reducing the amount of petrochemical raw materials in, for example, plastics [222,223,224]. Technical lignins have been successfully used in composites with different synthetic polymers, such as poly(propylene) [225], poly(ethylene terephthalate) [226,227], poly(vinyl alcohol) [228], poly(ethylene oxide) [229], poly(vinyl chloride) [230], polystyrene [231,232], polyethylene [233,234], and poly(lactic acid) [235]. The addition of lignin to other polymers not only decreases the amount of toxic and not renewable materials, but it can also improve the properties of the composites. The flame retardant properties of lignins have been used to increase the thermal stability of different composites such as polypropylene [236], poly(ethylene terephthalate) [226], poly(butylene adipate-co-terephthalate) [237], or thermoplastic starch [163].

One of the industries that is utilizing lignins as a filler is the rubber industry. Rubber needs reinforcing fillers for a proper performance of the material. One of the main products in this industry are tires, which, due to their stiffness and hardness, need to incorporate fillers that improve abrasion, tearing, cutting, and rupture resistance [238]. Usually, carbon black is the most common filler utilized in the tire industry, which is a petrochemical material that is non-degradable, with a great energetic demand and pollution in its production [239]. Technical lignin is a perfect green alternative that already has been studied, enhancing the properties and lowering the price of the rubber products they are added to [240,241]. Sodium lignosulfonate has been recently used as a way to improve the interfacial adhesion of latex and wool fabric, as well as the density and UV resistance of the resulting composites [242]. Nevertheless, since lignin is polar and rubber is nonpolar, there are difficulties in the compatibility of both materials [243], generating a variety of studies for the incorporation of lignin in the rubber matrix. The simplest approach is to blend the lignin with the rubber matrix; although, the new compounds show low compatibility and dispersability, as previously mentioned [241,243]. More sophisticated methods involve the application of high temperature dynamic heat treatment (HTDHT), improving the dispersion of the lignin in the matrix and even enhancing the properties of the composite [244]. A different approach is the latex co-precipitation method, in which the lignin is first mixed with latex and, afterwards, the rubber additives are added [245,246]. Technical lignin can also be modified to increase the hydrophobicity and, therefore, its compatibility with the rubber. There are different compounds that have been successfully used to improve the compatibility of lignin and rubber, such as (triethoxysilyl)propyl isocyanate (Figure 11) [247], benzoyl peroxide [248], or cyclohexylamine [249].

A different industry in whose products technical lignins could be used as filler is in inks, varnishes, and paints. Using residue lignin in black ink formulations has been known since the last century, both as itself and with chemical modifications [250,251,252,253]. Three-dimensional printing is a state-of-the art technology that would potentially have a huge impact in the near future since it can be used for rapidly creating structures for biomedical applications, functional materials, or energy storage systems [254,255,256]. These techniques tend to use petroleum-based plastics, such as epoxy resins [257], that should be replaced by greener alternatives, such as lignin. Other natural polymers, such as cellulose, have been tried as an alternative, but they have proven to be very vulnerable to humidity and ultraviolet-radiation aging, making the structures produced very vulnerable to the environmental elements [258].

As has been described previously in the chapter, there are plenty of applications for side-stream lignin, some of them having been known for decades [206,233,250,252]. Nevertheless, new applications for enzymatically modified technical lignins are currently being researched.

## 6. Environmental Assessment

Over the past few decades, the European Union and many other countries across the globe have been enacting policies and programs to encourage a shift from fossil fuels-based to renewable-based industry, with the European Green Deal aiming to improve the well-being and health of citizens and future generations [38]. The life cycle assessment (LCA) is a powerful tool in conducting a comprehensive environmental assessment for a product. It evaluates the impacts generated by a product from its every stage of production to its disposal. With the number of proposed lignin applications increasing, it is necessary to ensure pathways that have the least environmental impacts before commercialization [3,20].

Due to the novelty of enzyme-modified lignin upgraded to higher value products, there have been very few LCA studies investigating the environmental impacts of these products. The majority of environmental assessment studies investigate the use of unmodified lignin in the production of chemicals, including phenolic compounds and fuel [2,3,259] or the use of technical lignins as fillers or additives to replace other materials. For example, the life cycle assessment of the use of lignin-based binders to replace the binders of medium-density fiberboards was thoroughly investigated by Yuan et al. [21]. The extent of using lignin as a substitute for petroleum-based components, however, may also entail property and mechanical challenges. Thus, a change in the composition of a final product such as a composite may be needed when lignin-based components are used to ensure equal functionally with full petroleum-based compositions [3]. Hermansson et al. could show for the LignoBoost process in kraft lignin that the use of refined technical lignins has a significant influence on the climate impact of products with negative impacts ranging from −20% to several −100%, depending on the substitution level and allocation [23]. In a 2021 study, the environmental impact of the lignin production process was shown, meaning that a thorough assessment of the pulping process is vital [260]. Weiß et al. produced an enzymatically polymerized lignosulfonates coating for soil improvers, of which an LCA was performed with regard to the CO_2_ impact. It could be shown that enzymatically modified lignosulfonates hold great potential in terms of saving greenhouse gas emissions, as significant CO_2_ savings for the coating when compared to fossil based coatings were achieved [199].

## 7. Challenges and Opportunities

(1)Many factors influence the properties of enzymatically modified lignins. Aside from the extraction process and the type of biomass (softwood, hardwood, or grasses) used, the severity of the extraction process also affects lignin reactivity. A longer cooking time during kraft pulping is known to lead to higher condensed structures (formation of unreactive C-C linkages), meaning a lower reactivity. Thus, one needs to consider the characteristics of the respective lignin chosen for further planning. Thorough characterization of the used lignin is necessary to determine its reactivity and the direct comparison to other studies is difficult, since differences in analyses between different studies may occur and many factors affect lignin reactivity [121].(2)Harsh conditions as challenge for enzymes. Although a lot of different technical lignins are available nowadays, their uses in biocatalyzed processes are still limited. One reason is the insolubility of most technical lignins in water, making the use of organic solvents or alkaline solutions necessary where enzymes show limited stability. Another reason is the fact that industrial processes mostly are operated at elevated temperatures. Unfortunately, under these conditions, most of the fungal laccases are not active anymore. Therefore, a possible solution can be the use of thermostable enzymes, such as laccases from thermophilic organisms or such of bacterial origin, with the latter especially known to be more robust. Thus, they also would be active at the process conditions found in the industry. However, the problem with bacterial laccases is that they are not as well-characterized as fungal laccases by now, and that cultivation and purification can be rather challenging resulting in only low yields of pure enzyme. Another approach is the genetic modification of fungal enzymes to achieve high activities at higher temperatures or in organic solvents [7,9].(3)Controlling polymerization and de-polymerization processes. One problem related to thermal and chemical lignin depolymerization is the occurrence of uncontrolled re-polymerization of the generated smaller lignin fragments. One possibility to solve this problem could be the chemical conversion of the reactive intermediates to more stable chemical compounds. Furthermore, the reduction in reaction times would be useful to prevent repolymerization. Another solution would be the physical removal of the reactive intermediates from the reaction mixture. It also would be possible to use enzymes since they are highly selective. Thereby, lignin can be treated either directly with enzyme mixtures or indirectly with whole microorganisms. The enzymes and microorganisms used can also be selected or modified further to obtain the best results [121,261].(4)Markets and environmental impact of enzymatically produced lignin-based products. Since many of the lignin valorization technologies are emerging and novel, the production costs could be much higher than the conventional products. Lignin-derived products can play an important role in increasing our reliance on renewable-based chemicals, fuels, and materials, and reducing the carbon footprint of products and processes. Therefore, one factor that could be considered for future scale up of the valorization processes is the size of the potential market (or multiple markets) and the price volatility of the targeted product. To ensure that commercialization is performed in a safe, responsible, and sustainable way, the environmental impacts of the commercialized product should be investigated.

## 8. Conclusions

To tackle public issues such as global warming and pollution, organizations and investors must adjust not only the politics but the socio-economic framework as well. Enhanced global cooperation might boost technical lignin commercialization by channeling funds and facilitating the discovery of innovative routes. In the coming years, research is expected to investigate the usage of lignin in a variety of advanced next-generation applications, as is evident by the current European funding scheme. In our review, we introduced emerging studies on enzymatically modifying technical lignin into mechanical responsive hydrogels coatings, adhesives, and fillers. It is foreseen that more lignin-based smart materials with high performance will be designed and prepared in the future. Moreover, the increasing number of enzymatically modified technical lignins with flexible physicochemical properties allow for more advanced techniques in lignin processing. Lignin-based polymeric materials could be promising alternatives to traditional fossil-based materials. Given the progress of lignin modification chemistry and the development of new processing techniques and techniques for technical lignin purification, lignin is a promising renewable resource for high performance materials. Lignin valorization has emerged as an important research field in the near future.

## Figures and Tables

**Figure 1 polymers-15-01694-f001:**
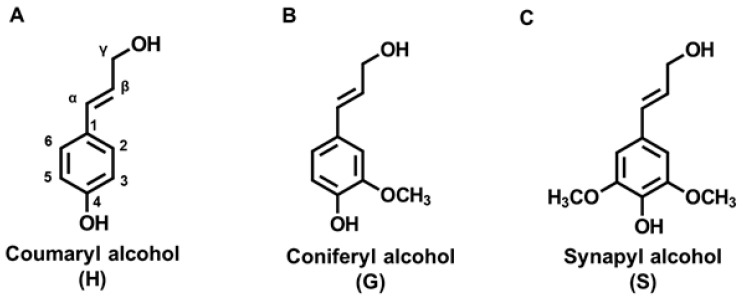
Structure of the monolignol unit precursors of lignins. Coumaryl- (**A**), coniferyl- (**B**) and sinapyl-alcohol (**C**).

**Figure 2 polymers-15-01694-f002:**
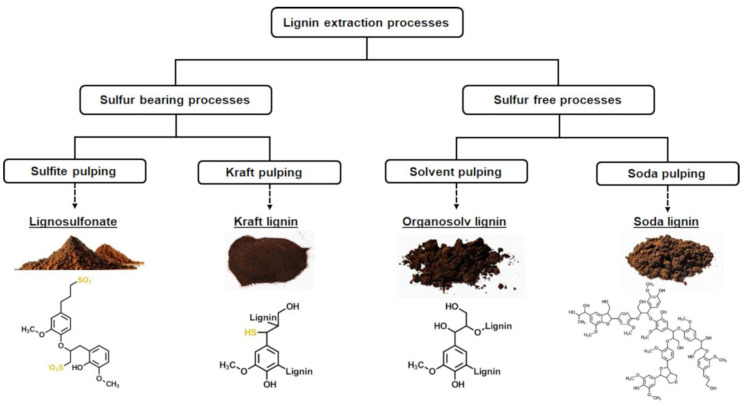
Overview of the most common lignin extraction processes and representative structures of the generated technical lignins.

**Figure 3 polymers-15-01694-f003:**
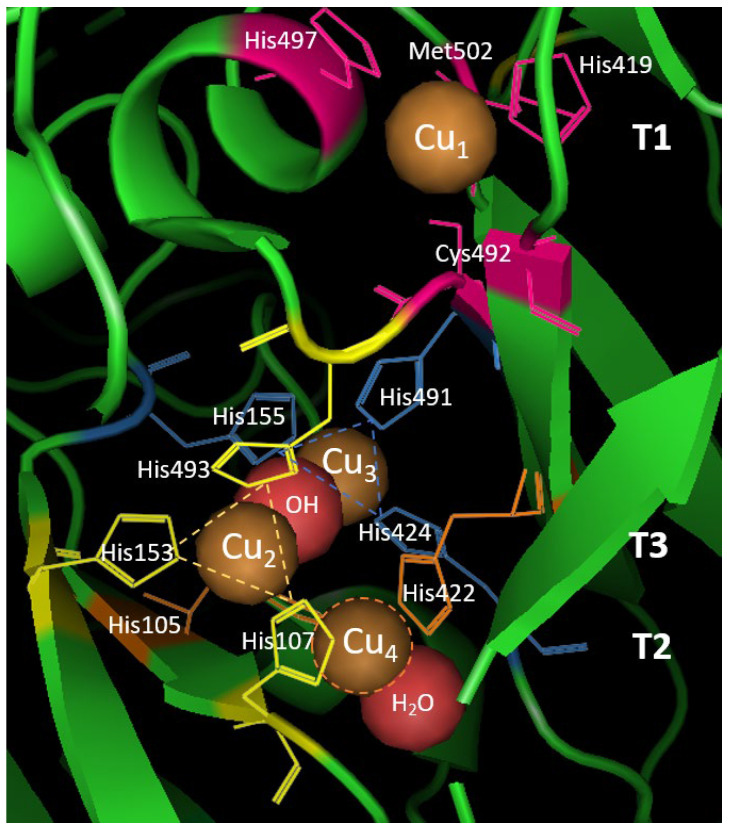
Detail of CotA laccase active site from *Bacillus subtilis*. In pink, the four ammino acids that coordinate with the Type1 copper; in yellow and blue, the three histidines residues that coordinate with Type3 copper atoms; in orange, the two histidines that bind with Type2 copper atoms. PDB code: 1gsk.

**Figure 4 polymers-15-01694-f004:**
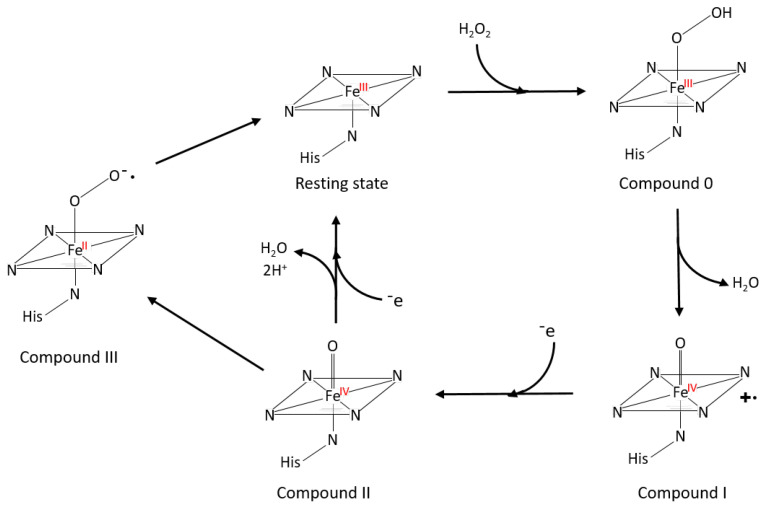
General catalytic cycle of heme peroxidases.

**Figure 5 polymers-15-01694-f005:**
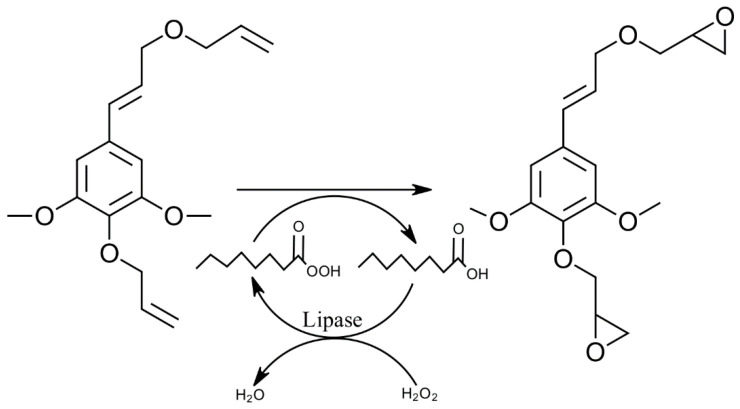
Epoxidation of alkylated lignin (represented as sinapyl alcohol) using a lipase, hydrogen peroxide, and caprylic acid.

**Figure 6 polymers-15-01694-f006:**
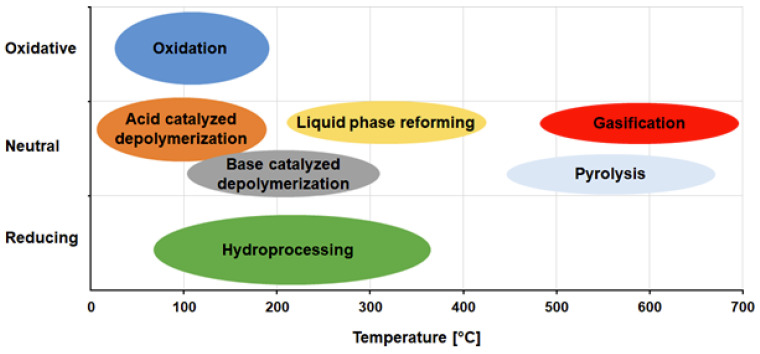
Overview of the chemical and thermal lignin conversion processes (modified from Li et al., 2015).

**Figure 7 polymers-15-01694-f007:**
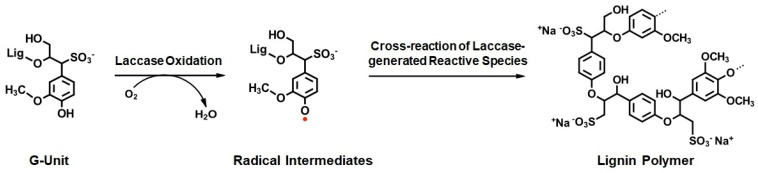
Reaction mechanism of laccase catalyzed polymerization of lignosulfonate [150].

**Figure 8 polymers-15-01694-f008:**
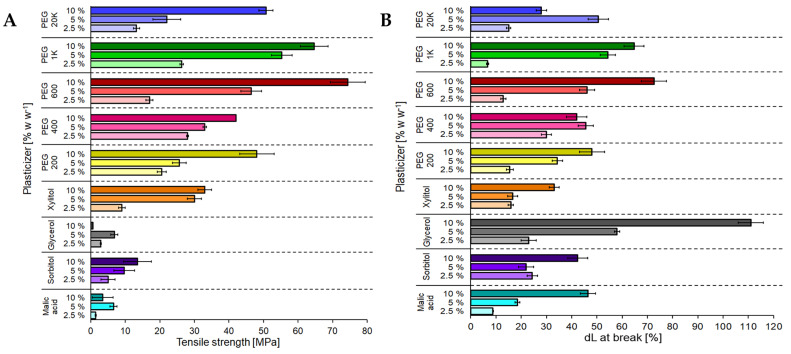
Effects of different plasticizers, in various amounts, on the tensile strength (**A**) and elongation at break (**B**) of laccase-polymerized lignosulphonates [154].

**Figure 9 polymers-15-01694-f009:**
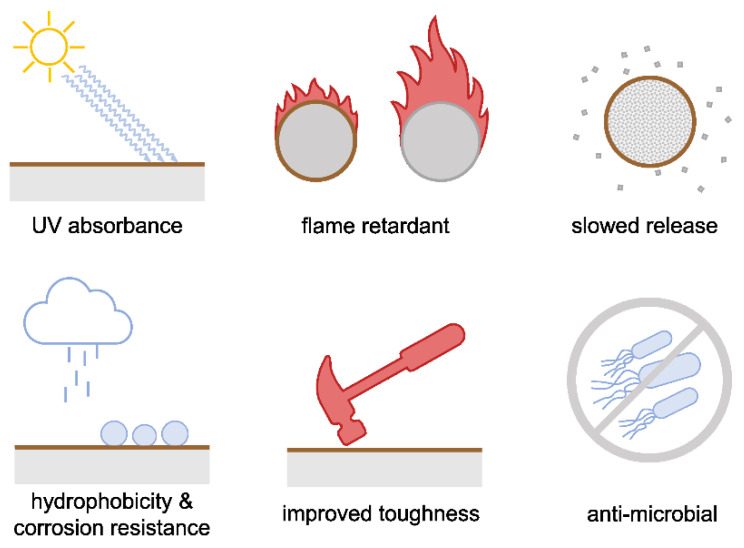
Lignin coatings can provide improved toughness and hydrophobicity, corrosion resistance, anti-microbial and flame-retardant properties, protection against UV-radiation, and slowed release of substrate materials.

**Figure 10 polymers-15-01694-f010:**
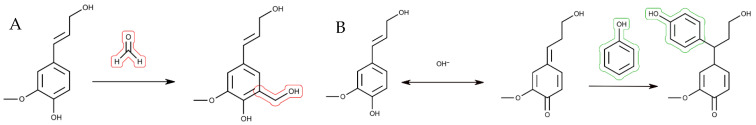
Proposed reaction mechanisms of lignin (represented as guaiacyl alcohol) with formaldehyde (**A**) and phenol (**B**).

**Figure 11 polymers-15-01694-f011:**
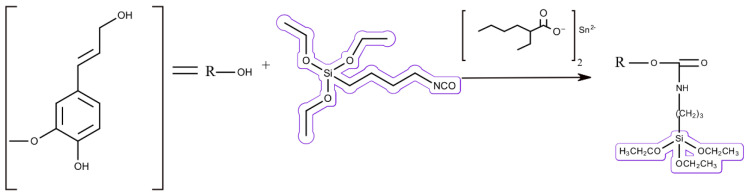
Reaction mechanism of lignin with (triethoxysilyl)propyl isocyanate in order to improve the combability with rubber (adapted from [247]).

## Data Availability

Not applicable.

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
