# Peer review of "The Biomodified Lignin Platform: A Review"

_polymers, 2023, doi:10.3390/polym15071694_

Round 1

Reviewer 1 Report

The presented review "The Biomodified Lignin Platform: A Review" includes more information than indicated in the title. Its content is fully consistent with Polymers and is likely to be in high demand due to the relevance of such research. The strengths of the review include the authors' attempt to present all types of industrial lignins and a description of the methods of biotechnological treatment of them to obtain valuable products. It is known that lignins exist in very different structures, therefore, in this case, information is given for lignins that are obtained (as a by-product of the pulp industry) in the European part. The review covers 238 sources, but for the review in 2023 it needs to be enriched with fresh references. It is possible that the authors would prefer to exclude old references. For publication, it is necessary to clarify some details and eliminate shortcomings.

Weaknesses to be fixed:

1. A very beautiful title of the review, and the abstract distracts the reader with general ecological arguments. It is necessary to more accurately express the main idea about the value of lignin as an independent platform of demanded substances.

2. In the enumeration of methods for obtaining native lignin, one could indicate a reference to the hydrotropic method of wood processing (the use of sodium benzoate).

3. You need to include information and cite the following works:

- Polym. Chem. 2015. 6. 4497. DOI: 10.1039/c5py00263j

- Zhang et al. Carbon Research (2022) 1:14. https://doi.org/10.1007/s44246-022-00009-1

- Renewable and Sustainable Energy Reviews 154 (2022) 111822. https://doi.org/10.1016/j.rser.2021.111822.

4. Add more articles from 2022 and 2023 to the list of publications

5. Check the list of publications for correct formatting and their accessibility for the reader.

Author Response

Dear Reviewer,
Thank you very much for the valuable notes on improving our manuscript. Please find a point by point response on your concerns below and in the manuscript in blue color.
Fabbri et al had a view on the recent progress made in enzymatically modifying technical lignins utilizing laccases, peroxidases and lipases. The study is complete and comprehensive. However, some minor comments are needed to improve.
Please re-writ the abstract. It is not complete and not include all sections of an abstract.
The abstract has been rewritten accordingly.
The introduction section is so long. It must be shorten.
We have shortened the manuscript accordingly omitting sections about general information on native lignin.
The manuscript needs a brief methodology section: the data sources, the search method,.....
A methodology section was added to the manuscript as chapter 2. Methodology.
The quality of some figures is low: 7 and 8.
Both figures were changed to higher quality.
Generally, this review has been written good. I enjoy it. All sections are complete. The main point is being long in all sections!
It can be accepted after improving the comments.

Reviewer 2 Report

Fabbri et al had a view on the recent progress made in enzymatically modifying technical lignins utilizing laccases, peroxidases and lipases. The study is complete and comprehensive. However, some minor comments are needed to improve.

Please re-writ the abstract. It is not complete and not include all sections of an abstract.

The introduction section is so long. It must be shorten.

The manuscript needs a brief methodology section: the data sources, the search method,.....

The quality of some figures is low: 7 and 8.

Generally, this review has been written good. I enjoy it. All sections are complete. The main point is being long in all sections!

It can be accepted after improving the comments.

Author Response

Dear Reviewer,

Thank you very much for the valuable notes on improving our manuscript. Please find a point by point response on your concerns below and in the manuscript in blue color.

The presented review "The Biomodified Lignin Platform: A Review" includes more information than indicated in the title. Its content is fully consistent with Polymers and is likely to be in high demand due to the relevance of such research. The strengths of the review include the authors' attempt to present all types of industrial lignins and a description of the methods of biotechnological treatment of them to obtain valuable products. It is known that lignins exist in very different structures, therefore, in this case, information is given for lignins that are obtained (as a by-product of the pulp industry) in the European part. The review covers 238 sources, but for the review in 2023 it needs to be enriched with fresh references. It is possible that the authors would prefer to exclude old references. For publication, it is necessary to clarify some details and eliminate shortcomings.

Weaknesses to be fixed:

  1. A very beautiful title of the review, and the abstract distracts the reader with general ecological arguments. It is necessary to more accurately express the main idea about the value of lignin as an independent platform of demanded substances.

Abstract has been reworked accordingly.

  1. In the enumeration of methods for obtaining native lignin, one could indicate a reference to the hydrotropic method of wood processing (the use of sodium benzoate).

Thank you for this comment. We have added a sentence and references accordingly in the manuscript.

  1. You need to include information and cite the following works:

- Polym. Chem. 2015. 6. 4497. DOI: 10.1039/c5py00263j

- Zhang et al. Carbon Research (2022) 1:14. https://doi.org/10.1007/s44246-022-00009-1

- Renewable and Sustainable Energy Reviews 154 (2022) 111822. https://doi.org/10.1016/j.rser.2021.111822.

The above mentioned publications were added to the references and put in context with the manuscript.

  1. Add more articles from 2022 and 2023 to the list of publications

The references were updated accordingly.

  1. Check the list of publications for correct formatting and their accessibility for the reader.

References have been checked and modified in Mendely to be visualized correctly.
